# High Propagule Pressure and Patchy Biotic Resistance Control the Local Invasion Process of the Tree *Ligustrum lucidum* in a Subtropical Forest of Uruguay

**DOI:** 10.3390/plants14060873

**Published:** 2025-03-11

**Authors:** Alejandro Brazeiro, Federico Haretche, Carolina Toranza, Alexandra Cravino

**Affiliations:** 1Grupo Biodiversidad y Ecología de la Conservación, Instituto de Ecología y Ciencias Ambientales, Facultad de Ciencias, Universidad de la República, Montevideo 11400, Uruguay; fharetche@gmail.com (F.H.); alecravino@gmail.com (A.C.); 2Departamento Forestal, Facultad de Agronomía, Universidad de la República, Montevideo 12900, Uruguay

**Keywords:** biological invasions, ecosystem resistance, glossy privet, hemiparasitism, propagule pressure

## Abstract

The tree *Ligustrum lucidum* (W. T. Aiton, Oleaceae), native to East Asia (China), has become an aggressive invader of subtropical and temperate forests around the world. To understand how its local small-scale spread is controlled, we studied (48 plots of 4 m^−2^), in a subtropical forest of Uruguay, the distribution and survival of seedlings, saplings, and poles to assess the effects of dispersal from mother trees (distance), microsite type (forest stands defined by dominant species), and past control measures. The propagule pressure of *L. lucidum*, estimated through seedlings density, was between 100 and 1000 times higher than that of other species of the community and was concentrated around mother trees (<10 m of distance). Spatial variability of seedlings, saplings, and poles densities were explained by the interaction between distance to mother trees and forest stands. Significative lower densities were observed in the forest patches (stands) dominated by *Jodina rhombifolia*, and a field survival experiment confirmed lower survival of poles at *Jodina* stands, demonstrating that some resistance mechanism is operating there. We propose two biotic mechanisms of resistance: herbaceous competition and/or roots hemiparasitism by *J. rhombifolia*, reducing seedling and sapling survival. We concluded that a high propagule pressure, small-scale dispersal from mother trees, and patchy biotic resistance at *Jodina* stands control the local spread and domination process of the tree *L. lucidum* in the studied forest.

## 1. Introduction

The invasion of ecosystems by exotic plant species is a global phenomenon that generates serious impacts on ecological, economic and social systems [1,2,3,4]. Understanding what factors, processes, and mechanisms regulate invasive processes and their effects on ecosystems is crucial to designing effective control and restoration programs.

In plant invasions, seedling recruitment represents a major filter to the spread of a species in a new habitat and is often limited by seed rain and microsite availability, but also by biotic interactions, both negative and positive [5]. The success of an invasive process depends on the balance between two opposite forces that vary in space and time: the propagule pressure, which promotes invasion, and ecosystem resistance, which tends to limit or reduce it [6,7]. “Propagule pressure” is defined as the number of individuals (or any organism parts capable of becoming an individual, like plant seeds or animal eggs) introduced into a new site and is calculated as the number of introduction events by the number of individuals per event [6,7]. The concept of “ecosystem resistance” was introduced early in the ecological literature to describe the forces, abiotic and biotic, that hinder the establishment of an alien species in a new site by affecting its growth, survivorship, or reproduction [8]. Ecosystem resistance represents the other side of the coin of what we call “invasibility”, defined as the susceptibility of a given local ecosystem to being invaded by a certain exotic species [8].

The tree *Ligustrum lucidum* (W. T. Aiton, Oleaceae) or glossy privet, native to East Asia (China), is a dangerous invasive species of subtropical and temperate forests around the world [9,10]. It is currently present on all continents except Antarctica [9], has expanded its invasion range into humid tropical areas [11], and its potential global distribution could reach 14,201,846 km^2^, an area slightly smaller than the size of South America [10].

*Ligustrum lucidum* has several life history traits that explain its high invasiveness [12], such as its high seed production [13], its fleshy fruits, attractive to birds, that favor the dispersal of its seeds [14], and its rapid growth in height, both in the shade and in direct sunlight, which allows it to dominate the forest canopy [15]. In advanced stages of invasion, this evergreen species forms almost monospecific patches, dominates the canopy, and generates low light conditions in the understory, hindering the establishment and growth of most species of trees, shrubs, lianas, and epiphytes, ultimately resulting in the reduction and even exclusion of several native plant species [16,17,18,19,20]. Glossy privet invasion has also consequences for animal diversity (e.g., soil invertebrates, birds) and ecosystem services (e.g., water provision) [9,21,22].

In Uruguay, glossy privet is one of the main threats for forest conservation, since it has become established in 13.4% of the forest lands; however, for the moment, it has only come to dominate the forest canopy, displacing native trees, in 1.2% [23]. The invasion of Uruguayan forests by *L. lucidum* is in the spread and impact stages, currently in expansion from the original focus localized in the S-SW region of the country [23]. In our opinion, the strategy for managing the invasion of privet in Uruguayan forests should focus, on the one hand, on preventing invasion in those forests not colonized by glossy privet through monitoring and early warning. On the other hand, the strategy should promote the control or containment of the invasion in forests not yet dominated, especially those that are relevant for conservation and that have human resources to carry out management measures. Our research aims to contribute to this last type of management recommendation, which could apply to the approximately 10% of Uruguayan forests which are invaded but no dominated yet.

To understand how the local small-scale spread of *L. lucidum* is controlled, we studied here its recruitment in an invaded but not yet dominated forest in Uruguay - Melilla´s forest (Figure 1) with the aim of evaluating its propagule pressure and dispersal and detecting possible factors that generate ecosystem resistance. Three main questions are addressed: (1) How important is the propagule pressure of *L. lucidum* with respect to the native trees? (2) How does *L. lucidum* recruitment vary in space with respect to mother trees? (3) Does the invasibility of forests to *L. lucidum* depend on the type of forest stand, defined by the dominant tree species?

We first describe the overall seedlings, saplings, and poles densities of *L. lucidum* and other exotic and native trees in the forest. In this study, the density of seedlings was used as a proxy of propagule pressure. Second, we assess the effects of forest stands and distance to mother trees on *L. lucidum* regeneration (seedlings, saplings, and poles). Third, we analyze the effect of forest stands on the survival of *L. lucidum* poles. Finally, the effect of previous control activities (tree felling, 10 years ago) on current recruitment was analyzed.

## 2. Results

### 2.1. Regenerating Species Assemblage and L. lucidum Propagule Pressure

We identified 16 tree species in the regenerating assemblage of Melilla’s forest, half of them being exotic (Table 1). *L. lucidum* had the highest densities, which, at the seedling stage, were between 100 and 1000 times higher than the other species, both exotic and native. Among native species, *Blepharocalyx salicifolius* had the highest densities, but this was more than 200 times lower than *L. lucidum* at the seedling stage and about 10 times lower at the sapling and poles stages.

These results clearly indicate that *L. lucidum* propagule pressure in Melilla’s forest is between one and two orders of magnitude higher than the other species, both exotic and native (Figure 2). The densities of *L. lucidum* seedlings, saplings, and poles were higher than the respective densities of the sum of the rest of the exotics species, as well as the sum of all native species (GLM binomial, *p* < 0.0001) (Figure 2).

According to our sampling, *L. lucidum* represented 99% of the seedling assemblage and approximately 88% of the saplings and poles assemblages, including all native trees and the rest of the exotics.

The survival of *L. lucidum* described a typical type III survival curve, with maximum mortality during the early stages (seedlings to saplings) that decreases in more mature stages (saplings to poles) (Figure 2). Assuming stability in recruitment and mortality rates, we estimated that only 6.1% of seedlings survive to become a sapling, and only 2.2% survive to the stage of pole. Despite the high mortality experienced by seedlings, the density of *L. lucidum* individuals reaching the pole stage exceeds the density of all other plants combined, both native and exotic (Figure 2).

The seedling dispersal of *L. lucidum* presented a very localized pattern around closer mother trees, fitting a logarithmic decay with increasing distance (Figure 3). Most seedlings were recruited within 10 m of the nearest mother trees, whereas the occurrence of seedlings at largest distances was relatively rare. The effect of seed source was also evident at the stage of sapling, since a logarithmic decay with distance to closer mother tree was observed (Figure 3). At the pole stage, the effect of the seed source faded, since the linear trend of density reduction with distance was not statistically significant (Figure 3). However, this effect became significant when the forest stand was incorporated in the model (see below, Section 2.2).

### 2.2. Exploring Determinant of L. lucidum Recruitment

The spatial variability observed in the densities of seedlings (coefficient of variability: CV = 245%), saplings (CV = 196%), and poles (CV = 182%) of *L. lucidum* was very high. The seedlings’ variability was explained by the effects of stand type and primarily by the distance to closer mother trees in interaction with stand type (Table 2). Seedling density decreased rapidly with distance to the mother tree under *Ligustrum* and *Scutia* stands: up to 2–3 m from mother trees, the density is in the order of hundreds to thousands of individuals per square meter, between 3 and 10 m distance, the density drops from hundreds to tens, and at greater distances, the density becomes almost constant, with few individuals per square meter (Figure 4). However, in *Jodina* stands, the density of seedlings was one or two orders of magnitude lower than in *Ligustrum* and *Scutia* stands and was relatively independent from the distance to the mother tree (Figure 4).

In the case of saplings, the spatial variability was explained by stand type and distance to closer mother trees, and no significant interactions between these variables were detected (Table 2). Saplings density showed a similar pattern to that of seedlings, but an order of magnitude lower. Density decreased rapidly with distance and was highest in *Ligustrum* stands, intermediate under *Scutia*, and lower under *Jodina* stands (Figure 4).

Finally, poles density was also affected by distance to mother trees and stand type (Table 2), showing a density pattern comparable to the previous stages, with values one order of magnitude lower (tens of individuals) than the seedlings (Figure 4). Density of poles decreased with distance to mother tree, with higher values in *Ligustrum* and *Scutia* stands than in *Jodina* stands (Figure 4).

### 2.3. Survival Experiment: Assessing Stand Type Effects

In addition to the expected pattern of offspring’s dispersion around mother trees, we detected a clear spatial pattern of reduced *L. lucidum* regeneration in *Jodina* stands (Figure 4b). Regardless of proximity to mother trees, the density of seedlings, saplings, and poles was much lower in *Jodina* stands than in *Ligustrum* stands but was also lower than in *Scutia*. This pattern suggests that *L. lucidum* regenerants recruited in *Jodina* stands have lower survival. To test this hypothesis, we conducted a natural experiment to track the survival of nearly a thousand poles naturally recruited in different stand types over 1–3 years.

The best-fitted logistic model included significant effects of stand type and poles height, while time did not have a significant effect (Table 3). Survival was highest in *Ligustrum* stands (mean: 0.98, IC95%: 0.96–1.0), intermediate in *Scutia* (mean: 0.86, IC95%: 0.80–0.92), and lowest in *Jodina* (Mean: 0.75, IC95%: 0.66–0.84), and tended to increase slightly with the height of poles (Table 3, Figure 5).

### 2.4. Possible Underlying Factors Behind Stand Type Effects

Several physiognomic and environmental differences were detected among the stand types we studied here (Figure 6). The stands of *Ligustrum*, *Scutia*, and Control (ten years before) presented closed canopies, with covers of 85–95%, contrasting with the relative open *Jodina* stands (60–70% cover) (Figure 6a). This difference was reflected in the proportion of light reaching the understory, with very low transmittance in the stands of *Ligustrum* (mean ± SD: 0.6 ± 0.4%) and *Scutia* (4.9 ± 3.8%), and higher in *Jodina* stands (23.0 ± 11.6%), with similar values to relative open sites where *L. lucidum* had been recently (2 years before) controlled (Figure 6b). The shrub covers were in general low, around 1–2% in all stand types, except in *Scutia* stands, where there was about 5% (not presented data). However, herb cover showed an important variability among stand types, with high values in *Jodina* stands (60%) and low values (10–25%) in the rest of the stands (Figure 6c). In *Jodina* stands, the herb layer was always dominated by *Melica sarmentosa* Ness, a native perennial climbing herbaceous of the Poaceae family. Forest height was low in general, slightly higher in *Ligustrum* stands (7 m) than in *Jodina* ones (6 m) (Figure 6d). Tree regeneration cover (from seedlings to poles) was high (50%) in the controlled stands (i.e., tree fell 10 years ago) and relatively low (5–10%) in the rest of the stands (Figure 6e), while litter cover was similar in all stands, varying around 40–50% (Figure 6f).

### 2.5. Effects of Previous Control Activities on Current Recruitment of L. lucidum

We compared the regeneration density of *L. lucidum* (seedlings, saplings and poles) in *Ligustrum* stands (i.e., invasion without control) with controlled stands, both 2 and 10 years before our sampling, to assess the effects of adult tree control on posterior regeneration success. The best-fitted models for seedlings, saplings, and poles included the effects of control type (C2, C10, *Ligustrum*) and distance to closer mother tree in interaction with control type (Appendix A). We found that seedling density was strongly reduced (80–90% on average) in controlled stands in comparison with uncontrolled stands, especially in old controls (10 years old) (Figure 7). Saplings density was also strongly reduced (89% on average) in stands controlled 10 years ago, with respect to *Ligustrum* stands, but in stands controlled 2 years ago, the pattern was reversed, with a density that on average was more than twice that observed in unmanaged stands (Figure 7). At the stage of poles, the highest density was registered in the controlled stands, seven to ten times greater than in *Ligustrum* stands (Figure 7).

## 3. Discussion

### 3.1. High Propagule Pressure of L. lucidum

Propagule pressure is a key factor of plant invasion [7], and it has been quantitatively recognized as a significant characteristic of invasion in trees [24,25]. Indeed, it is currently recognized that propagule pressure needs to be integrated as a basis of a null model when studying the process of tree invasion [26].

How important is the propagule pressure of *L. lucidum* with respect to the native trees? We used here the density of seedlings as proxy for the propagule pressure of *L. lucidum* in Melilla’s forest. The estimated mean seedling density was 253.7 ind.m^−2^, with a maximum of 1200 ind.m^−2^, which agrees with global antecedents of 200 to >600 per m^2^ in the invaded areas [9]. Sixteen regenerating tree species were identified in Melilla’s forest, of which half were exotic. *Ligustrum lucidum* had by far the highest seedling density, which was between 2 and 3 orders of magnitude higher than the other species, whether natives or exotics. The second most important species in the seedling community was *B. salicifolius*, a very common native tree, which had a density 250 times lower than *L. lucidum*. This result clearly shows the very high propagule pressure of *L. lucidum* within the seedling community of Melilla’s forest, which undoubtedly plays a central role in the success of its invasion. The predominance of *L. lucidum* propagule pressure over the native tree flora has been previously documented, both at the stages of seedlings, e.g., [14], and seeds, produced by trees and stored in soils, e.g., [15,27].

The high seed production, which could be as high as 100.000–10.000.000 seeds per tree and per year, the high germination rate (70–95%) and wide environmental tolerance [9] explain the high propagule pressure of *L. lucidum*. However, experimental evidence produced in Brazil indicates that germination rate is high when seeds are released at short stage after fruit abscission, but low rates of germination were observed in all stored seeds, indicating low vigor related to loss and possibly to consumption of reserve material for embryo support during the 20 days of storage [28]. Thus, the propagule pressure and therefore the invasive potential of *L. lucidum* seems to be more associated with the great number of seeds produced than with their germination potential (rusticity).

### 3.2. Small-Scale Dispersal from Parental Trees

How does the *L. lucidum* recruitment vary in space with respect to the mother trees? We found that most seedling dispersal exhibited a highly localized pattern around mother trees, fitting a decreasing logarithmic pattern with increasing distance. Most seedlings were recruited within 10 m of the nearest mother plants, whereas the occurrence of seedlings at large distances was relatively rare. Surely most of the seedlings recruited within a 10 m radius from mother trees come from seeds directly fallen from trees (i.e., seed rain).

However, bird-mediated dispersal of seeds can also be of short distance inside dense forests. Powel and Aráoz [29] analyzed bird-mediated dispersal of *L. lucidum* seeds in a secondary forest and surrounding crop-fields in the Yungas biome (northwestern Argentina) and found that seeds are transported shorter distances in dense forest than in less dense areas. They concluded that *L. lucidum* seeds are dispersed low distances inside the forest because tree density reduces dispersal distances by birds, suggesting that the invasion that occurs within the forest can be delayed in dense forests. It should be noted that in Melilla’s forest, we have estimated an average density of 2094 trees.ha^−1^ (DBH ≥ 10 cm) (not published data), somewhat higher (1200 trees.ha^−1^, DBH ≥ 10 cm) than that documented in the study of Powel and Aráoz [29], so the mechanism they proposed could also be applicable for our study forest.

While long-distance dispersal mediated by bird frugivory is a relevant regional process for the invasion (colonization) of new localities [14,15,29], at a local scale, nearby dispersal mainly by seed rain seems to play a central role in the process of local spread and domination. This implies that within a certain temporal frame, some type of dispersal limitation may occur, even for invasive trees such as *L. lucidum*, providing some opportunities for the management of the invasion process.

The signal of the seed source effect on seedlings density was also evident at the stage of saplings (on average 1–2 years old), since their density also exhibited a logarithmic decrease with distance to the closest mother trees. In poles (on average 2–5 years old), the signal of the seed source effect was not clear when the data of all stand types were analyzed together, but it was evident when *Ligustrum* and *Scutia* stands were considered separately.

These results are consistent with the relative low spread velocity observed in the field in Cordoba (Argentina), with values ranging between 11 and 12.5 m.yr^−1^, slightly lower than the 13.69 m.yr^−1^ predicted by a model [30]. Discrepancies between simulation and field values are due to the effects of habitat and biotic interactions [30].

### 3.3. Biotic Resistance in Jodina Stands to L. lucidum Invasion

We asked whether ecosystem resistance to *L. lucidum* invasion varies among different forest stands. We found dramatic differences between the very high and high densities of seedlings, saplings, and poles observed in *Ligustrum* and *Scutia* stands compared to the low densities registered in *Jodina* stands. At the same distance to the nearest mother trees, and therefore probably receiving comparable seed-rains, seedling recruitment was drastically lower in *Jodina* stands. The same pattern was observed at the stages of saplings and poles, suggesting that the survival of *L. lucidum* is reduced by the ecological conditions of *Jodina* stands. By tracking the survival of almost a thousand poles naturally recruited in different stand types over 1–3 years, we demonstrated that survival in *Jodina* stands (75%) was lower than that estimated in *Scutia* and *Ligustrum* stands (86–98%). These results show that some process of ecosystem resistance operates in the *Jodina* stands, which translates into a reduction in the seedling recruitment rate and a higher mortality rate of saplings and poles. This is the first time, to our knowledge, that evidence of biotic resistance exerted by *J. rhombifolia* on glossy privet regeneration has been documented. We do not know whether this pattern is limited to the Melilla Forest or has some degree of generality in the region or in other forest types. Therefore, considering the possible implications of this finding for the management of *L. lucidum* invasion in forests, it is very important to determine how generalizable it is. In this sense, we have detected another similar case (i.e., reduced recruitment of glossy privet in *Jodina* stands) in a hillside forest in eastern Uruguay (*Pers. Comm*. Cravino 2020).

We propose two non-exclusive hypotheses to explain the ecosystem resistance in *Jodina* stands: the first one proposes interference from herbaceous vegetation as the mechanism. Mazia et al. [31] carried out field experiments sowing seeds of exotics trees in grasslands and found a strong limitation in seedlings emergence in the case of *L. lucidum* and important seedling mortality in other three woody exotic species. In fact, the addition of seed to grassland did not produce any seedlings of *L. lucidum* in their assays. Finally, they suggested that the spread of exotic woody plants, including *L. lucidum*, into remnants of mesic pampean grassland can be strongly reduced by interference from resident vegetation. Based on these antecedents, generated in the same region and biome of our study area, we propose that herbaceous vegetation, which reaches high coverage in *Jodina* stands, reduces seedling recruitment and the survival of saplings and poles via competition. The herbaceous cover in *Jodina* stands (60%) is twice that in *Scutia*, and it is six times greater than in *Ligustrum*. This is explained because the foliage architecture of *J. rhombifolia* trees generates relatively open canopies, allowing greater entry of light into the understory. The native grass *Melica sarmentosa*, a typical pioneer species of forest gaps, dominates the herbaceous layer in *Jodina* stands. This species is therefore the main candidate to further evaluate the competition hypothesis between herbaceous plants and *L. lucidum.*

Our second hypothesis proposes the mechanism of roots hemiparasitism by *J. rhombifolia*. An interesting feature of *J. rhombifolia* is that it is a hemiparasite tree, capable of removing water and mineral nutrients from host plants through haustorial connections with roots, as was documented in the case of *C. tala* and *S. buxifolia* in Argentina [32]. Root hemiparasitic plants both compete with and extract resources from host plants; they are generalists and attach to and parasitize a wide variety of hosts simultaneously, mainly the dominant hosts [33]. It was suggested that generalist native adversaries, such as hemiparasitic plants, may impede the success of invaders due to the lack of defense or tolerance mechanisms of the host plants against parasitism [34]. In this context, using cultivation experiments, it was demonstrated that native root hemiparasites can effectively decrease alien clonal plants’ biomass production and shoot density [35]. Based on this evidence, although mostly associated with herbaceous plants, we propose that through root competition and parasitism, *J. rhombifolia* reduces the survival of seedlings, saplings, and poles of *L. lucidum.*

### 3.4. Isolated Control of Adult Trees Is Not Sufficient to Restore Forests Invaded by L. lucidum

The previous control of *L. lucidum* adult trees (i.e., cutting and application of herbicide on stumps) carried out in the area 2 and 10 years before our sampling has generated undesirable changes in the forest. Although the control strongly reduced the present-day density of seedlings due to the reduction of seed sources, it generated higher densities of saplings (2 times) and mainly of poles (7–10 times). The cutting of adult trees generated at that moment an opening of the canopy, which surely allowed more light to reach the forest floor, stimulating the survival and growth of *L. lucidum* seedlings. Currently, the stands controlled 10 years before our sampling are severely invaded by poles and adult trees of *L. lucidum*, the species dominating the canopy, while the coverage of native species is minimal. Therefore, the control activity carried out, focused exclusively on the removal of adult trees, clearly failed to restore the invaded stands.

This experience shows the importance of assessing the success of control methods of tree invasions not only in the short-term with respect to the mortality of adult trees but also in terms of the medium-term probability of re-invasion, assessing the survival and growth of saplings. The experience also demonstrates that seedlings and saplings must also be considered in the control strategy of this invasive tree. After killing adult trees, saplings and poles should be removed annually until the seed bank is depleted. Otherwise, the management could generate counterproductive results.

## 4. Materials and Methods

### 4.1. Study Area

The Melilla Forest is included in the national protected area “Humedales del Santa Lucía”, in the department of Montevideo, in southern Uruguay (Figure 1, upper panel). It is a subtropical mesophilic forest, distributed along a sedimentary ravine separating high-dry areas (10 m isobath) with grasslands, shrublands, and woodlands from low-humid areas (2 m isobath) dominated by wetlands (Figure 1, mid panel). The average temperature in the area is 17.3 °C, varying between a minimum of 11 °C in July (winter) and a maximum of 23 °C in January (summer). Annual rainfall is 1142 mm, which is evenly distributed throughout the year, but with strong interannual variability [36].

We studied a forest area of about 12 ha, with variable width (80–150 m) and approximately one kilometer in length (Figure 1, mid panel). It is a dense (canopy cover > 75%), relatively low (6–9 m), and diverse forest (26 native tree species). It has been closed to livestock since 2000. In 1983, a floristic study was carried out in this forest area, which did not report any exotic tree species [37], but 21 years later, a thesis documented the presence of *L. lucidum* and other exotic woody plants [38]. Nowadays, the forest canopy is dominated by three native species, in decreasing order, *Scutia buxifolia* Reiss, *Celtis tala* Planch and *Jodina rhombifolia* Hook. et Arn., but the fourth position is occupied by the exotic tree *L. lucidum* (*not published data*). In addition to glossy privet, other exotic woody plants have been observed, such as *L. sinense* Lour., *Laurus nobilis* L., and *Cotoneaster* sp.

### 4.2. Field Sampling

To assess the effects of forest physiognomy on the recruitment of *L. lucidum*, and other exotic and native trees, in 2016–2017, we surveyed 48 plots (2 × 2 m) randomly assigned to four stand types (Figure 1, mid and lower panels): (1) *Scutia* stand: closed canopy dominated by *Scutia buxiflora* and understory with low herbaceous cover. (2) *Jodina* stand: relative open canopy dominated by *Jodina rhombifolia* and understory with high herbaceous cover. Its distribution is patchy, with patches normally small (5–10 m radius), made up of a few adult trees of *J. rhombifolia*, plus some *Celtis tala* and *Scutia fuxiflora* trees. (3) *Ligustrum* stand: very closed canopy dominated by *L. lucidum* and understory without herbaceous cover. (4) Control stand: areas where adult trees of *L. lucidum* were felled in 2007–2008, ten years before our survey.

Seedlings (height: <10 cm), saplings (height: 10–50 cm) and, poles (height: >50 cm and diameter at breast height: DBH ≤ 2.5 cm) of exotic (privet and others) and native tree species were identified and quantified at each plot. In our experience with *L. lucidum* in Uruguay, on average, seedlings are less than 1 year old, saplings are 1–2 years old, and poles 2–5 years old, or older in some cases. The distances from the center of each plot to the three closest mother trees (i.e., flowering plants with DBH ≥ 10 cm) were measured to calculate the mean distance.

We also characterized physiognomy at every sampling plot by assessing the cover of bare soil, litter, woody debris, and vegetal layers (herbs, shrubs, tree regeneration, canopy trees, and emerging trees). We visually estimated vegetation cover by layer. The mean height of each vegetal layer was also measured. Seven plots per stand type (28 in total) were randomly selected to assess the light level in the understory using a multiple-point light sensor (ACCUPAR LP-80). At midday (13–14:30 h), we measured the Photosynthetically Active Range (PAR) (wavelengths between 400 and 700 nm) at each plot (three measurements were made in different directions to obtain a mean value) and in close open areas to estimate the transmittance (i.e., the ratio of PAR reaching the sample point to PAR measured in the open). We also took PAR measurements in stands where adult *L. lucidum* trees were logged in 2015, two years before our survey.

Between 2017 and 2018, fifteen circular plots of 5 m radius were delimited, three for each stand type, i.e., *Ligustrum*, *Scutia*, and *Jodina*, and all poles (height: >50 cm and DBH ≤ 2.5 cm) of *L. lucidum* were marked and measured (maximum height). The survival (survivor = 1, dead = 0) of marked individuals was registered in two or three opportunities between 2019 and 2021. A total of 999 individuals were followed between one year (379 days) to more than three years (1333 days).

### 4.3. Data Analysis

The densities of *L. lucidum* seedlings, saplings, and poles were compared with the corresponding densities of all exotic species together, and with the densities of all native species together, using a Generalized Linear Model (GLM) with a binomial distribution after detecting overdispersion problems (i.e., mean residual variance much higher than 1) when using Poisson models.

The spatial variability of the density of *L. lucidum* seedlings, saplings, and poles was modeled using GLM with Poisson distribution and log link function. The models assessed the effects of two variables, distance to mother trees (continuous variable) and stand type (categorical variable with three levels: *Scutia*, *Jodina*, and *Ligustrum* stands), and the interaction between them (distance–stand). Both distance and stand type were modeled as fixed variables (glm: Y ~ Distance + Stand + Distance: Stand, family = Poisson).

To assess the effect of the previous control of *L. lucidum* (i.e., felling of mature trees and herbicide application on stumps) on current recruitment success, we modelled the current densities of seedlings, saplings and poles in invaded non-managed stands (i.e., *Ligustrum* stands) and managed ones (i.e., control stands), both 2 and 10 years ago, using GLM with Poisson distribution.

The models included the fixed effects of two variables, distance to mother trees and stand type (categorical variable with three levels: *Ligustrum*, Control 2 years ago, and Control 10 years ago), and the interaction between them (distance–stand) (glm: Y ~ Distance + Stand + Distance: Stand, family = Poisson).

To search for evidence of ecosystem resistance to *L. lucidum* invasion in the different stands, we analyzed the survival data (survivor = 1, dead = 0) from 999 poles tracked for 1 to 3 years. The main effect to assess was stand type, contrasting *Ligustrum*, *Scutia*, and *Jodina* stands. But the survival of poles may also be dependent on the individual height and on the elapsed time, expecting reduced survival rates in smaller plants and over longer elapsed times. Thus, we used GLM to fit logistic regressions with binomial distributions and logit link function (i.e., ln(p/1–p)) to model *L. lucidum* (poles) survival in function of three fixed factors, stand type (categorical variable with three levels: *Ligustrum*, *Scutia*, and *Jodina*), initial height of individuals (in cm), and elapsed time (days) (glm: Poles Survival ~ Stand + Heigh + Time, family = binomial).

Different models were estimated for each case, including null models with only intercepts and models of increasing complexity, from additive (only direct effects of predictors) to multiplicative (with interactions). The selection of the best models was carried out by retaining predictors with significant effects (*p* < 0.05) and choosing the most parsimonious ones according to the Akaike Information Criteria (i.e., lower AIC, at least 2 units), among those models meeting the assumptions of homogeneity and independence of residuals according to visual checks (using classic plots). When necessary, post hoc comparisons between stand types were conducted using Fisher’s LSD test. All analyses were performed in RStudio, R version 4.4.2 [39].

## 5. Conclusions

Understanding the factors that control the spread and dominance of an invasive alien species in a natural ecosystem is crucial for projecting potential impacts on biodiversity and designing effective management strategies. Seedling recruitment represents an important filter for the spread of invasive trees, a process that is usually controlled by the interaction between seed dispersal and the quality and availability of microsites. We concluded here that a high propagule pressure (i.e., seedlings density), small-scale dispersal from mother trees, and patchy biotic resistance in *Jodina* stands control the local small-scale spread of *L. lucidum* in Melilla’s forest.

In terms of learning to manage the *L. lucidum* invasion, our results showed that controlling adult trees in isolation is not a recommended method, as it favors the survival and growth of saplings and therefore the reinvasion of the controlled sites. Saplings and poles should also be considered in the management strategy and should be eliminated annually until the seedling and seed banks are exhausted. To reduce the probability of reinvasion in the future by improving biotic resistance, forest management should also promote the persistence and greater abundance of *J. rhombifolia*. Finally, we conclude that controlling adult trees in isolation, leaving aside seedlings and saplings, is not a recommended method since it improves the light conditions that favor the growth of saplings and thus reinvasion.

## Figures and Tables

**Figure 1 plants-14-00873-f001:**
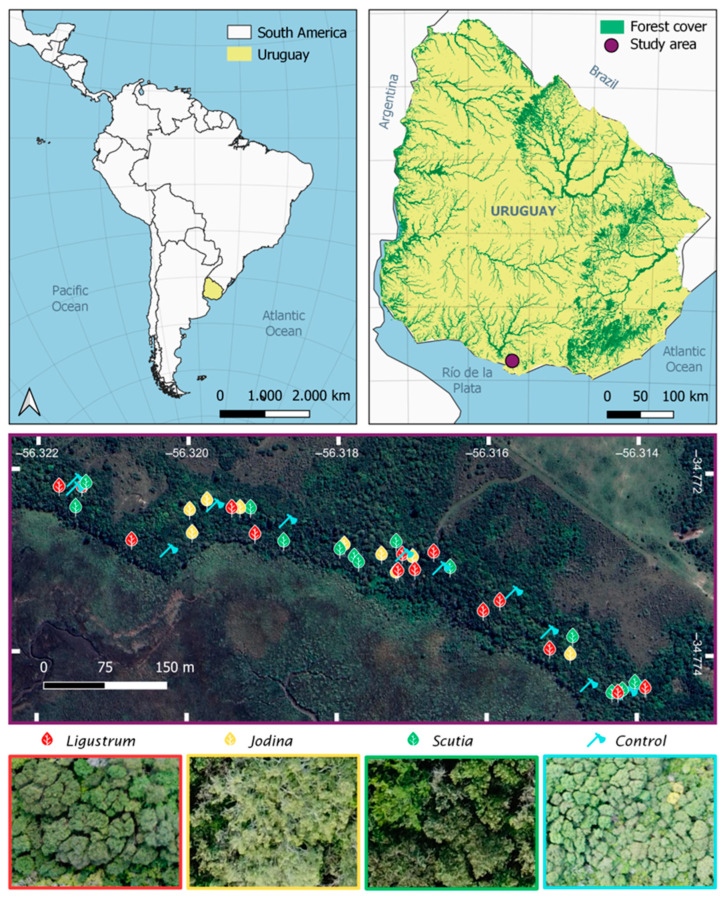
Study area (Melilla´s forest) and field sampling plots distribution. Upper panel: Uruguay location and Uruguayan forest cover. Mid panel: Distribution of 48 sampling plots (2 × 2 m) randomly allocated into four stand types, depending on the canopy dominant tree species: *Ligustrum lucidum*, *Jodina rhombifolia*, *Scutia buxifolia,* and control sites (*L. lucidum* adult trees fell 10 years ago). Lower panel: Canopy photographs of each type of stand: *Ligustrum*, *Jodina*, *Scutia* and Control.

**Figure 2 plants-14-00873-f002:**
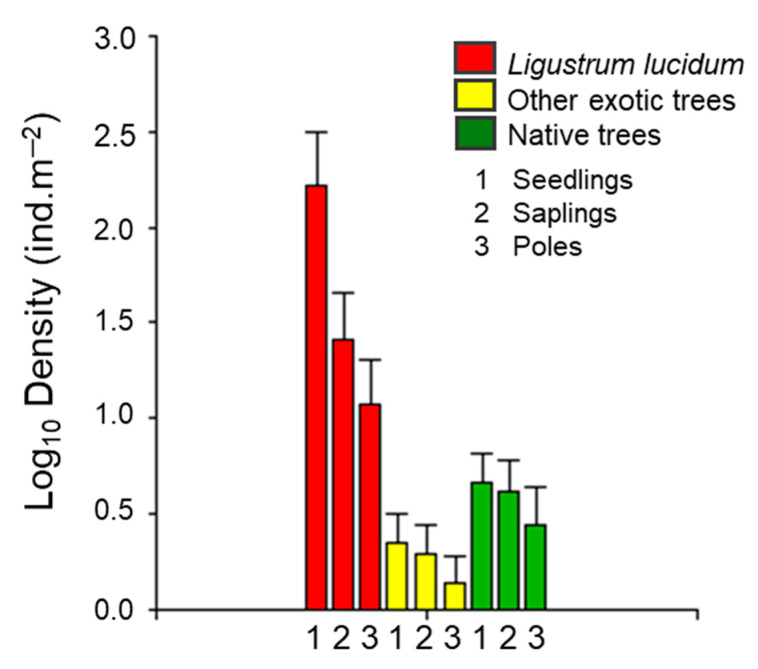
Regeneration density of *L. lucidum*, other exotics, and native tree species in Melilla’s forest. Overall mean values (log10 scale) of seedlings, saplings, and poles. Bars indicate confidence intervals of 95% (CI95%). Statistical differences between *L. lucidum* and other exotic and native trees were detected by binomial GLMs (*p* < 0.0001), in seedlings, saplings, and poles.

**Figure 3 plants-14-00873-f003:**
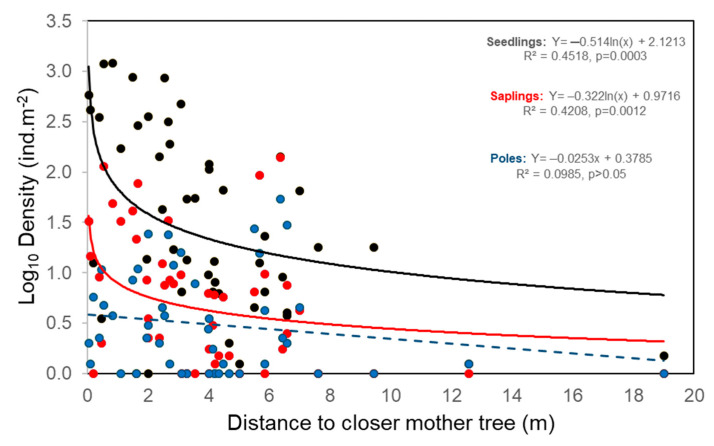
General dispersal pattern of *L. lucidum* in Melilla’s forest. Seedling, sapling, and poles densities (log10 scale) variation in relation to distance to closer mother trees. Data from all plots (n = 48), including the four stand types studied (*Jodina*, *Ligustrum*, *Scutia*, Control), were included in the analysis. The best-fitted models are presented (equations and solid lines).

**Figure 4 plants-14-00873-f004:**
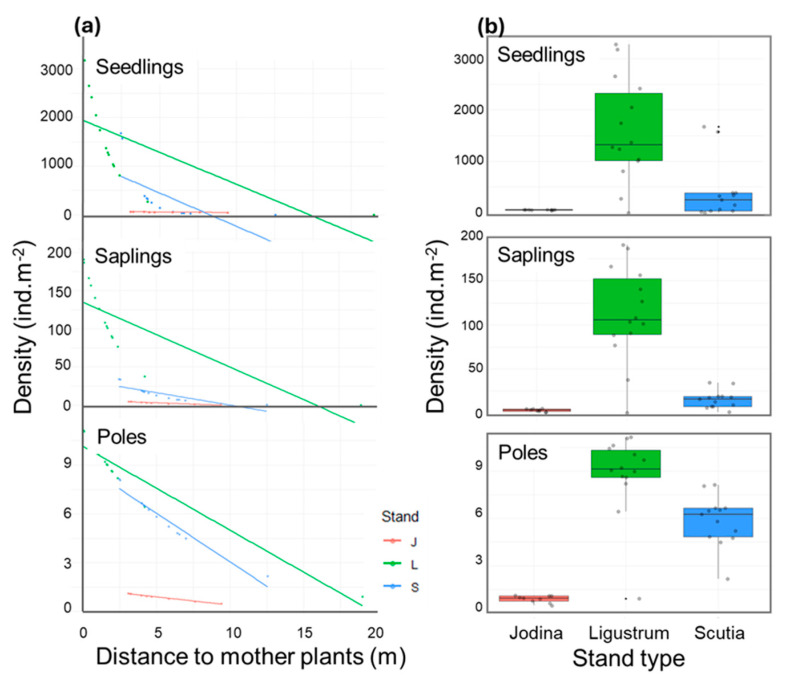
Effects of distance to mother trees and stand type (J: *Jodina*, L: *Ligustrum*, S: *Scutia*) on the densities of *Ligustrum lucidum* seedlings, saplings and poles in Melilla’s forest. Predictors of the best GLM models fitted are given (see Table 2). (**a**) Combined effects of distance and stand type. (**b**) Global effect of stand type represented in box plots.

**Figure 5 plants-14-00873-f005:**
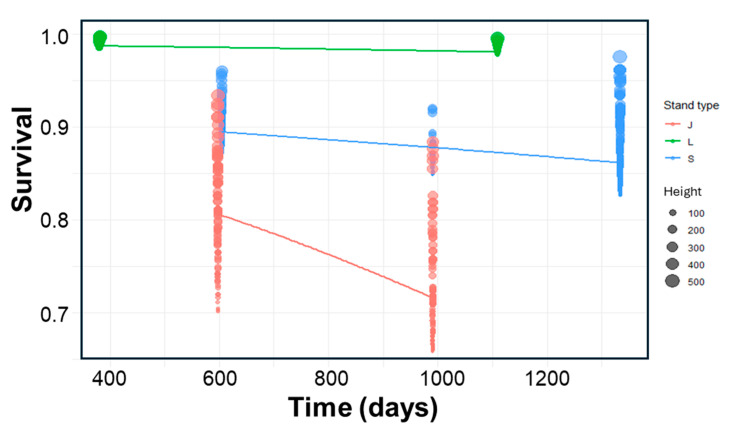
Effects of stand type, individual height, and time on the survival of *Ligustrum lucidum* poles in Melilla’s forest. Predictions of the best-fitted model (logit GLM) are given (see Table 3).

**Figure 6 plants-14-00873-f006:**
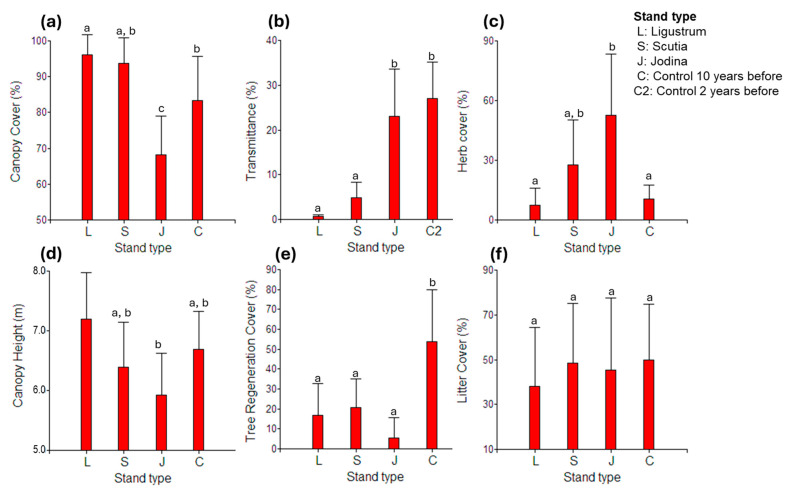
Physiognomic characterization of the forest stands analyzed. Mean values and confidence intervals (95%) are presented for the main measured descriptors: (**a**) canopy cover, (**b**) light transmittance, (**c**) herb cover, (**d**) canopy height, (**e**) tree regeneration cover, and (**f**) litter cover. Different letters indicate significant differences (*p* < 0.05) according to ANOVA and post hoc tests.

**Figure 7 plants-14-00873-f007:**
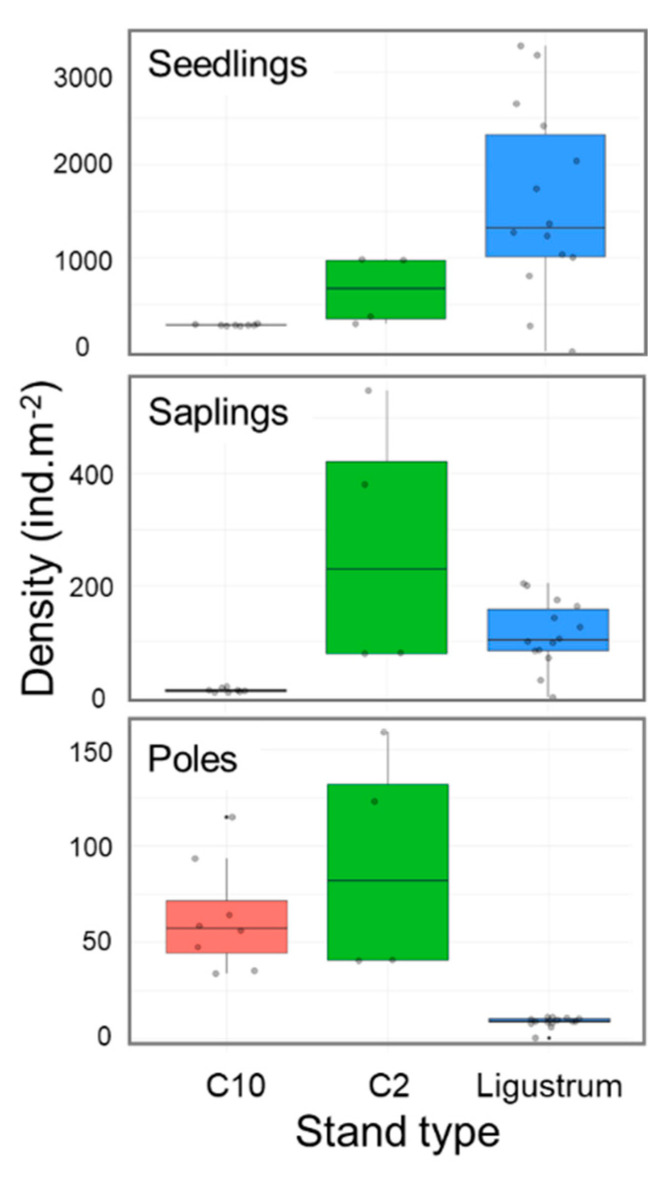
Effects of control activities on the densities of *Ligustrum lucidum* seedlings, saplings, and poles in Melilla’s forest. Predictions of the best-fitted GLM models are given (see Appendix A) using box plots.

**Table 1 plants-14-00873-t001:** List of regenerating trees in Melilla’s forest. Mean density and standard deviation (SD) of seedlings, saplings, and poles are given. Exotic and native species are indicated.

	Density (ind.m^−2^) (SD)		
	Seedlings	Saplings	Poles
Exotic species			
*Ligustrum lucidum* W.T. Aiton	253.7 (621.4)	15.5 (30.3)	5.6 (10.2)
*Ligustrum sinense* Lour.	0.2 (0.3)	0.05 (0.16)	0.03 (0.10)
*Laurus nobilis L.*	0.03 (0.1)	0.23 (0.5)	0.06 (0.2)
*Cotoneaster* sp.	0	0.01 (0.1)	0.02 (0.1)
*Pyracantha coccinea* M. Roem	0	0.01 (0.0)	0.01 (0.0)
*Phoenix canariensis* H. Wildpret	0	0.01 (0.0)	0
*Morus alba* P.	0	0	0.01 (0.0)
*Pittosporum undulatum* Vent.	0	0.06 (0.4)	0.01 (0.0)
Native species			
*Blepharocalyx salicifolius* (Kunth) O. Berg	1.07 (1.47)	1.52 (2.10)	0.44 (1.62)
*Myrsine laetevirens* Mez	0.08 (0.36)	0.02 (0.08)	0.01 (0.04)
*Jodina rhombifolia* (Hook. & Arn.) Reissek	0.07 (0.26)	0.02 (0.08)	0.01 (0.05)
*Scutia buxifolia* Reissek	0.04 (0.09)	0.01 (0.05)	0.04 (0.15)
*Celtis tala* Gillies ex Planch.	0.04 (0.15)	0.01 (0.04)	0.01 (0.05)
*Acca sellowiana* (O. Berg) Burret	0	0.03 (0.18)	0.01 (0.04)
*Eugenia uniflora* L.	0	0.01 (0.04)	0.01 (0.04)
*Schinus longifolia* (Lindl.) Speg.	0	0.01 (0.05)	0

**Table 2 plants-14-00873-t002:** Best GLM models of density of *L. lucidum* seedlings, saplings and poles. The coefficient estimates for stand types were assessed with respect to *Jodina* stands. Significance codes: ‘***’ 0.001, ‘**’ 0.01, ‘*’ 0.05, ‘..’ 0.1, ‘NS’ > 0.1. The predictor names are as follows: Distance: distance to closer mother plants, Stand-L: *Ligustrum* stand, Stand-S: *Scutia* stand. *Jodina* stands were used as reference.

	Estimate	Std. Error	z Value	Pr(>|z|)	
Seedlings					
(Intercept)	4.21835	0.11929	35.36	<2 × 10^−16^	***
Distance	−0.04003	0.02262	−1.77	0.0768	
Stand-L	3.89434	0.11970	32.53	<2 × 10^−16^	***
Stand-S	5.56176	0.12813	43.41	<2 × 10^−16^	***
Distance-Stand-L	−0.55919	0.02410	−23.21	<2 × 10^−16^	***
Distance-Stand-S	−0.91487	0.02656	−34.45	<2 × 10^−16^	***
Saplings					
(Intercept)	2.7507	0.222	12.381	<2 × 10^−16^	***
Distance	−0.3899	0.0242	−16.109	<2 × 10^−16^	***
Stand-L	2.5125	0.2140	11.738	<2 × 10^−16^	***
Stand-S	1.7401	0.2122	8.199	2.43 × 10^−16^	***
Poles					
(Intercept)	0.518	0.3923	1.307	0.1912	
Distance	−0.1314	0.0372	−3.532	0.0004	***
Stand-L	1.9012	0.3836	4.957	7.17 × 10^−7^	***
Stand-S	1.9071	0.3717	5.131	2.89 × 10^−7^	***

**Table 3 plants-14-00873-t003:** Best model (logit GLM) of survival of *L. lucidum* poles. The coefficient estimates for stand types were assessed with respect to *Jodina* stands. Significance codes: ‘***’ 0.001, ‘**’ 0.01, ‘*’ 0.05, ‘NS’ > 0.1. The predictor names are as follows: Stand-L: *Ligustrum* stand, Stand-S: *Scutia* stand. *Jodina*’ stands were used as reference. Height: initial individual height (cm). Time: elapsed time (days).

	Estimate	Std. Error	z Value	Pr (>|z|)	
(Intercept)	0.9009358	0.4556859	1.977	0.04803	*
Stand-L	3.1828581	0.4531533	7.024	2.16 × 10^−12^	***
Stand-S	1.0696586	0.2555241	4.186	2.84 × 10^−5^	***
Height	0.0047637	0.0017515	2.720	0.00653	**
Time	−0.0004883	0.0004007	−1.219	0.22299	NS

## Data Availability

The datasets used during the current study are available from the corresponding author on reasonable request.

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
