# Peer review of "High Propagule Pressure and Patchy Biotic Resistance Control the Local Invasion Process of the Tree Ligustrum lucidum in a Subtropical Forest of Uruguay"

_plants, 2025, doi:10.3390/plants14060873_

Round 1
Reviewer 1 Report
Comments and Suggestions for Authors
The manuscript mainly evaluated the densities of seedlings, saplings and poles in relation to distance to mother L. lucidum in different forest stand. Results showed with the increasing distance to mother plants, the density of L. lucidum decreased, and the density and survival in Jodina stands were lowest. The manuscript is in general well-written, the research topic is intriguing, the conclusions are supported by the results, the references are appropriate. There are also some shortcomings that need to be addressed before publication. My main comments and suggestions are listed as follows.
From the title, we can see that this paper mainly explored the propagule pressure of invasive plants, and the Introduction also mentioned the calculation of propagule pressure (line 72), but it seems that in this paper there is no calculation of propagule pressure, but instead of measuring the density of L. lucidum.
Line 2, what does ‘patchy’ mean? I did not find the ‘patchy’ evidence in the paper.
Line 113-119, Using the present densities as a proxy for survival or mortality is a bit inaccurate, in case that invasive plants may be in a period of rapid population expansion with high seedling density.
Line 397, what about other plants coverage in the control stand after 2 or 10 years of logging?
Line 23, Except herbaceous competition, does it possible that the intraspecific competition between L. lucidum was also affect the density of L. lucidum?
Author Response
Comments 1: From the title, we can see that this paper mainly explored the propagule pressure of invasive plants, and the Introduction also mentioned the calculation of propagule pressure (line 72), but it seems that in this paper there is no calculation of propagule pressure, but instead of measuring the density of L. lucidum.
Response 1: We agree with Reviewer 1. The density of seedlings was used as a proxy of propagule pressure, but we did not make this explicit. So, we included in L90 the following sentence to better explain this point: In this study the density of seedlings was used as a proxy of propagule pressure.
Comments 2: Line 2, what does ‘patchy’ mean? I did not find the ‘patchy’evidence in the paper.
Response 2: The distribution of Jodina rhombifolia trees in the Melilla Forest is patchy. Patches are usually small (about 5-10 m radius) and are generally made up of a few large J. rhombifolia trees, accompanied by a few individuals of Scutia buxiflora and Celtis tala. To clarify this point, we modified: L21-22 as follow: “Significative lower densities were observed in the forest patches (stands) dominated by Jodina rhombifolia, ...”
L416-419 as follow: “(2) Jodina stand: relative open canopy dominated by J. rhombifolia and understory with high herbaceous cover. Its distribution is patchy, with patches normally small (5-10 m radius), made up of a few adult trees of J. rhombifolia, plus some Celtis tala and Scutia buxiflora trees.
Comments 3: Line 113-119: Using the present densities as a proxy for survival or mortality is a bit inaccurate, in case that invasive plants may be in a period of rapid population expansion with high seedling density.
Response 3: We agree, and for this reason we had explained in lines L115-116, that: “Assuming stability in recruitment and mortality rates, we estimated that only 6.1% of seedlings survive until become a sapling, and only 2.2 % survive to the stage of pole.”
Comments 4: Line 397, what about other plants coverage in the control stand after 2 or 10 years of logging?
Response 4: The coverage of other plants is currently minimal, while the density of poles and young adult trees of L. lucidum is very high. This was explained in L379-380:…..,while the coverage of native species is minimal.
Comments 5: Line 23, Except herbaceous competition, does it possible that the intraspecific competition between L. lucidum was also affect the density of L. lucidum?
Response 5: Yes, it is possible. In other not published study we found evidence of density-dependent mortality of saplings. But this could not be interpreted as biotic resistance under the frame of invasion ecology.
Reviewer 2 Report
Comments and Suggestions for Authors
The manuscript of Brazeiro et al. addresses an importante topic in the field of biological invasions. Its main aim is to understand the factors that drive the invasion of the invasive tree Ligustrum lucidum in a particular Uruguayan ecosystem. Overall, the manuscript is well written with only few issues marked in the attached file.
The title of the manuscript is in accordance with the objective defined in the abstract. However, at the end of the Introduction section, the authors mention analyzing the impact of control measures, but this is not included as an objective in the abstract. In my opinion, the addition of this objective in the abstract would improve the overall understanding of the manuscript.
In relation to the Materiial & Methods section, I recommend moving it to follow the Introduction section. I believe that this change will benefit the understanding of the manuscript. This change would would enhance clarity and coherence of the manuscript. The authors use Generelized Linear Models (GLM) to assess the effects of density and stand type on the survival of L. ligustrum. The selection of the best model is based on the use of Akaike Information Criterion (AIC), which is OK, but I think that the authors would provide more details about the models used in the analysis. This would provide rigor and a more comprehensive overview of the findings and the replication of the study.
The discussion of the manuscript is sufficiently deep but it would benefit from the inclusion of more references to previous studies. This would help to contextualize the findings and strenghten the overall quality of the manuscript. In addition, the authors should include the limitations of this study (for example, by referring that the study was conducted in a specific/ particular ecosystem, limiting its generability to other ecosystems.
Based on my comments and those provided in the attached file, I do not recommend its publication in its present form. I suggest that the authors do the necessary revisions to enhance its overall clarity and quality.

Author Response
REVIEWER 2
Comment 1: However, at the end of the Introduction section, the authors mention analyzing the impact of control measures, but this is not included as an objective in the abstract. In my opinion, the addition of this objective in the abstract would improve the overall understanding of the manuscript.
Response 1: We agree. We did not include this point (secondary objective) in the first manuscript due to limited space in the abstract. We now included this secondary objective in the abstract, modifying the lines 14 to 16 as follow: To understand how its local small-scale spread is controlled, we studied (48 plots of 4m-2) in a subtropical forest of Uruguay the distribution and survival of seedlings, saplings and poles, to assess the effects of dispersal from mother trees (distance), microsite type (forest stands defined by dominant species) and past control measures.
In lines 459-460 we include: Finally, we conclude that controlling adult trees in isolation, leaving aside seedlings and saplings, is not a recommended method, since it improves the light conditions that favor the growth of saplings and thus reinvasion.
Comment 2: In relation to the Material & Methods section, I recommend moving it to follow the Introduction section. I believe that this change will benefit the understanding of the manuscript. This change would enhance clarity and coherence of the manuscript.
Response 2: We agree. We organized the text of the first version using the template provided by the journal, which marked the location of the Materials and Methods section. If possible, for the Journal, we prefer to change the location of this section as indicated by the reviewer.
Comment 3: The authors use Generelized Linear Models (GLM) to assess the effects of density and stand type on the survival of L. ligustrum. The selection of the best model is based on the use of Akaike Information Criterion (AIC), which is OK, but I think that the authors would provide more details about the models used in the analysis. This would provide rigor and a more comprehensive overview of the findings and the replication of the study.
Response 3: We made the following changes in the text to explain in more detail the modelling process:
L428: Text inserted: Both distance and stand type were modeled as fixed variables (glm: Y ~ Distance + Stand + Distance:Stand, family = poisson).
L429-432: Text substituted by: To assess the effect of the previous control of L. lucidum (i.e., felling of mature trees and herbicide application on stumps) on current recruitment success, we modelled the current densities of seedlings, saplings and poles in invaded non-managed stands (i.e., Ligustrum stands) and managed ones (i.e., control stands), both 2 and 10 years ago, using GLM with Poisson distribution.
The models included the fixed effects of two variables, distance to mother trees and stand type (categorical variable with three levels: Ligustrum, Control 2 years ago and Control 10 years ago), and the interaction between them (distance-stand) (glm: Y ~ Distance + Stand + Distance:Stand, family = poisson).
L433-436: Text substituted by: To search for evidence of ecosystem resistance to L. lucidum invasion in the different stands, we analyzed the survival data (survivor=1, dead=0) from 999 poles, tracked for 1 to 3 years. The main effect to assess was stand type, contrasting Ligustrum, Scutia and Jodina´s stands. But the survival of poles may also be dependent on the individual height and on the elapsed time, expecting reduced survival rates in smaller plants size and longer elapsed times. Thus, we used GLM to fit logistic regressions with binomial distributions and logit link functions (i.e., ln(p/1–p)), to model L. lucidum (poles) survival in function of three fixed factors, stand type (categorical variable with three levels: Ligustrum, Scutia and Jodina), initial height of individuals (in cm) and elapsed time (days) (glm: Poles Survival ~ Stand + Heigh + Time, family = binomial).
L436-440: Text substituted by: Different models were estimated for each case, including null models with only intercepts and models of increasing complexity, from additive (only direct effects of predictors) to multiplicative (with interactions). The selection of the best models was carried out by retaining predictors with significant effects (p<0.05) and choosing the most parsimonious ones according to the Akaike Information Criteria (i.e., lower AIC, at least 2 units), among those models meeting the assumptions of homogeneity and independence of residuals, according to visual checks (using classic plots). When necessary, post hoc comparisons between stand types were done using Fisher´s LSD test. All analyses were performed in RStudio, R version 4.4.2.
Comment 4: The discussion of the manuscript is sufficiently deep, but it would benefit from the inclusion of more references to previous studies. This would help to contextualize the findings and strenghten the overall quality of the manuscript
Response 4: Suggestions for including new references were accepted (indicated in the new text).
Comment 5: In addition, the authors should include the limitations of this study (for example, by referring that the study was conducted in a specific/ particular ecosystem, limiting its generality to other ecosystems).
Response 5: The generalization of results derived from study cases is always problematic. This is a well-known limitation, so I believe it is not necessary to highlight this point oi general terms (full study), but it may be interesting for specific results. We addressed this point with respect to the possible biotic resistance exerted by J. rhombifolia.
We added this sentence in L318-325: This is the first time, to my knowledge, that evidence of biotic resistance exerted by J. rhombifolia on glossy privet regeneration has been documented. We do not know whether this pattern is limited to the Melilla Forest, or has some degree of generality in the region, or in other forest types. Therefore, considering the possible implications of this finding for the management of L. lucidum invasion in forests, it is very important to determine how generalizable it is. In this sense, we have detected another similar case (i.e., reduced recruitment of privet in Jodina stands) in a hillside forest in eastern Uruguay (Pers. Comm. Cravino 2020).
Comment 6 (inserted in MS): The methodology should be more explicit. You also have control plots and this fact is missed here. Futhermore, the "forest stand" should be explained because it represents different native species dominance.
Response 6: We agree. These suggestions have already been addressed in Response #1.
Comment 7 (inserted in MS): L38-39 Similar sentence in the Abstract. Please rewrite.
Response 7: Alternative text: …is currently a dangerous invasive species of subtropical and temperate forests around the world [5-6].
Comment 8 (inserted in MS): L49 and L52: It should be "Glossy privet". Please verify it through the manuscript
Response 8: OK, we use Glossy privet.
Comment 9 (inserted in MS): L52-54: Please, add a reference to support this sentence.
Response 9: The reference [19] (Brazeiro et al 2024) included in text support the complete sentence.
Comment 10 (inserted in MS): L67-69: Please, add a reference to support this sentence.
Response 10: This classic reference was added: [21] Williamson M (1996) Biological invasions. Chapman & Hall, New York
Comment 11 (inserted in MS): L84: What do you mean with "forest stands"? composition? density?
Response 11: We used forest stands, defined by dominant tree species, as microsite (important factor for tree recruitment).
Thus, we modified L83-84 as follows: (3) Does the invasibility to L. lucidum depend on the type of forest stand, defined by the dominant tree species?
Comment 12 (inserted in MS): Figure 4 Please see the graphs of Seedlings and Saplings. The blue and green lines are not in line with y-axis.
Response 12: The lines are predictions of the models, that include negative estimations in the cases indicated by the reviewer.
Comment 13 (inserted in MS): L187: There are different times in these observations. How could you do this analysis?
Response 13: Yes, due to the large number of plants marked, there was a certain delay in the monitoring of each plot. However, in each case the monitoring date was recorded, and the time elapsed in days between the different observations was calculated.
Comment 14 (inserted in MS): Table 2 and Table 3. Explain in detail the significance level used and predictors in the labels of the tables.
Response 14: Agree. Both labels were modified.
Comment 15 (inserted in MS): L208: This is new in the text. In the Material & Methods section, this is not mentioned. Please revise.
Response 15: Ok, we added in L433: We also took PAR measurements in stands where adult L. lucidum trees were logged in 2015, two years before our survey.
Comment 16(inserted in MS): The appearance of C2 introduces confusion in the understanding of the results. It appears one time, so what is the relevance of it?
Response 16: We consider this information to be relevant, as it allows us to observe how light conditions change (increase) immediately after the logging of adults, providing a logical explanation for the high densities of saplings and poles of L. lucidum observed 10 years after logging.
Comments 17 (inserted in MS): What is the relevance of this experiment here? In this study, you did not evaluate the germination of stored seeds. It will more important to refer the viability of seeds in the soil.
Response 17: The relevance of including this reference, is that provide relevant information to understand the high propagule pressure of Glossy Privet, what is resumed in L280-282: Thus, the propagule pressure and therefore the invasive potential of L. lucidum seems to be more associated to the great number of seeds produced, than to their germination potential (rusticity).
Comments 18 (inserted in MS): L272: The present study does not evaluate soil seed viability. So, the sentence is speculative.
Response 18: It is correct, we speculate (we used “seems to b”) but based on scientific information. Furthermore, in this part of the discussion we try to understand the high propagule pressure of glossy privet in general, and not specifically at our study site.
Comments 19 (inserted in MS): L283: Can you refer the density in this study, and compare with your results?
Response 19: According to our data, the Melilla´s forest has a somewhat higher tree density, so the mechanism suggested by Powell and Aráoz (2018) could make sense in our study site. We added in the manuscript that:
L296-299: It should be noted that in Melilla´s forest we have estimated an average density of 2094 trees.ha-1 (DBH≥10cm) (not published data), somewhat higher (1200 trees.ha-1, DBH≥10cm) than that documented in the study of Powel and Aráoz [29], so the mechanism they proposed could also be applicable for our study forest.
Comments 20 (inserted in MS): Your explanation seems correct. However, it is important to support your hypothesis with a reference.
Response 20: We used different references to support the premises of our hypothesis, but this new hypothesis is ours.
Comments 21 (inserted in MS): L367: Please see my previous comment on seed viability.
Response 21. L 387: we removed the part referred to seed viability.
Comments 22 (inserted in MS): L386-388: Please see your results. The most dominante native species is Blepharocalys salicifolius. So, whay do you select Jodina rhombifolia and Scutia buxifolia as the studied native species?
Response 22: This sentence refers to dominant canopy species, i.e., we are referring to adults trees, not seedlings, saplings or poles.
Comments 23 (inserted in MS): L400: The presence of other exotic/invasive species should be mentioned previously (Study area section).
Response 23: We agree.
L409-410, inserted text: In addition to glossy privet, other exotic woody plants have been observed, such as L. sinense Lour., Laurus nobilis L. and Cotoneaster sp.
Comments 24 (inserted in MS): L408: Here, what is the meaning of "emergent"?
Response 24: We change to “emerging trees”
Comments 25 (inserted in MS): Insert all authors in reference #3
Response 25: Done.
Reviewer 3 Report
Comments and Suggestions for Authors
Please attached pdf copy with annotations

Author Response
REVIEWER 3
Comments 1: 100-1000 is too high variation and needs statistics to look better and reliable.
Response 1: The abstract should be very brief to meet the format requirements of the journal. On the other hand, the differences in the level of recruitment are of 2 and 3 orders of magnitude, therefore, we believe that it is very clear to mark the differences in this way in the abstract, since the details are shown in Results.
Comments 2: what does 10 represent?
Response 2: 10 meters. We change the text: (<10m of distance).
Comments 3: use typical ecology terminology - I dont know that poles are and if they are alive?
Response 3: “Pole” is a juvenile state between a sapling and an adult. The term is very used in forestry ecology (See Newton A. Forest Ecology and Conservation; DOI: 10.1093/acprof:oso/9780198567448.001.0001).
Comments 4: herbaceous competition with what?
Response 4: With Ligustrum lucidum seedlings and saplings.
L26: text added: reduce seedling and sapling survival.
Comments 5: what does patchy biotic resistance mean? did you measure densities of native species also and found it patchy? If it is patchy, it means it is impaired and may allow invasion instead of defending the native patch – rephrase.
Response 5: This point was also raised by referee 2. The distribution of Jodina rhombifolia trees in the Melilla´s Forest is patchy. Patches are usually small (about 5-10 m radius) and are generally made up of a few large J. rhombifolia trees, accompanied by a few individuals of Scutia buxiflora and Celtis tala. To clarify this point, we modified L20-21 as follow: “Significative lower densities were observed in the forest patches (stands) dominated by Jodina rhombifolia, ...”; and L394-396 as follow: “(2) Jodina stand: relative open canopy dominated by J. rhombifolia and understory with high herbaceous cover. Its distribution is patchy, with patches normally small (5-10 m radius), made up of a few adult trees of J. rhombifolia, plus some Celtis tala and Scutia buxiflora trees.
Comment 6: too short introduction - I recommend you expand the first paragraph and describe and introduce the processes you will be investigating under invasion biology. This paragraph (L65-80) is misplaced, it can be moved to the begining of introduction.
Response 6: We agree. The paragraph (L65-80) was moved to the beginning of the Introduction, to expand the general presentation of the topic addressed.
Comment 7: L82-83: the questions 2 & 3 are similar and please seperate them in terms of what you investigated or rephrase based on data/methods or ...
Response 7: Question # 3 was reformulated as: 3) Does the invasibility to L. lucidum depend on the type of forest stand, defined by the dominant tree species?
Comment 8: your results must be percentages and counts in the 48 plots, and I thus, invite better statistical analysis
Response 8: The difference in density between glossy privet and the other tree species is so large and clear that, in our opinion, statistical comparison is not necessary. In any case, we used percentages in the following sentence (L113-115), which states that 99% of the individuals in the seedling assemblage are privets, and 88% in the case of saplings and poles.
Comment 9: L146: too high percentages (CV) and may mislead or means something not right . use linear mixed models to analyse the data.
Response 9: This can lead to problems when detecting patterns, and it is often suggested to use larger sampling units to reduce variance. In our case, however, we were able to find very clear spatial patterns (statistically significant), demonstrating that the evaluated predictors exert significant control over privet regeneration.
Comment 10: L 370: I am confused and stopped here. results come before methods? is this the journal style?
Response 10: We agree with the referee; it is not the usual structure. This order was stated in the template given in the journal web, for this reason we adopted it. But several recent papers published by Plants presented the standard structure. So, if it is possible for the journal, we prefer to use the classic format.
Round 2
Reviewer 1 Report
Comments and Suggestions for Authors
The author has made corresponding revisions based on my suggestions. I think the work is ready for publication.
Author Response
We would like to thank reviewer 1 for all her/his contributions, which have allowed us to improve our article.
Reviewer 2 Report
Comments and Suggestions for Authors
Dear authors,
I am satisfied with your work in this revised version. So, I recomment it for publication.
Author Response
We would like to thank reviewer 2 for all his/her contributions, which have allowed us to improve our article.
Reviewer 3 Report
Comments and Suggestions for Authors
I noticed the ignorance of my comments and wonder what it means. The editor will have to intervene as I reinstate the same comments in addition to the ones in the current revision (R1).

I noticed the ignorance of my comments and wonder what it means. The editor will have to intervene as I reinstate the same comments in addition to the ones in the current revision (R1).
Author Response
REVIEWER 3
Dear Reviewer 3,
We are sorry that you feel that your comments were ignored, as we took them seriously. Some suggestions were in fact accepted. In the case of suggestions that were not accepted due to space constraints, or to accommodate the suggestions of the other 2 reviewers, the decision taken in each case was justified. In your second revision, only your initial comments are indicated. We have not found any new comments on the second version. Therefore, we are replying to your first comments again, trying to be more precise and clear in the explanations.
Comments 1: 100-1000 is too high variation and needs statistics to look better and reliable.
Response 1: The abstract should be very brief to meet the format requirements of the journal. On the other hand, the differences in the level of recruitment are of 2 and 3 orders of magnitude, therefore, we believe that it is very clear to mark the differences in this way in the abstract, since the details are shown in Results.
We included in L114-116 (in green): The densities of L. lucidum seedlings, saplings and poles, were higher than the respective densities of the sum of the rest of exotics species, as well as the sum of all native species (GLM binomial, p<0.0001)) (Fig. 2).
Although the graphical data in Figure 2 were very clear in showing the highest densities of glosy privet, since the confidence intervals of the density estimators do not overlap, we added statistical analysis, binomial GLM, after detecting overdispersion problems when using a Poisson GLM. This addition was also explained in the Material and Methods section (L452-455).
Comments 2: what does 10 represent?
Response 2: 10 meters. We change the text: (<10m of distance).
Comments 3: use typical ecology terminology - I dont know that poles are and if they are alive?
Response 3: “Pole” is a juvenile state between a sapling and an adult. The term is very used in forestry ecology (See Newton A. Forest Ecology and Conservation; DOI: 10.1093/acprof:oso/9780198567448.001.0001).
We clearly defined the term "pole" in the section Material and Methods, L426-427: poles (height: >50 cm and diameter at breast high: DBH ≤ 2,5 cm).
Comments 4: herbaceous competition with what?
Response 4: With Ligustrum lucidum seedlings and saplings.
L26: text added: reduce seedling and sapling survival.
Comments 5: what does patchy biotic resistance mean? did you measure densities of native species also and found it patchy? If it is patchy, it means it is impaired and may allow invasion instead of defending the native patch – rephrase.
Response 5: This point was also raised by referee 2. The distribution of Jodina rhombifolia trees in the Melilla´s Forest is patchy. Patches are usually small (about 5-10 m radius) and are generally made up of a few large J. rhombifolia trees, accompanied by a few individuals of Scutia buxiflora and Celtis tala. To clarify this point, we modified L20-21 as follow: “Significative lower densities were observed in the forest patches (stands) dominated by Jodina rhombifolia, ...”; and L394-396 as follow: “(2) Jodina stand: relative open canopy dominated by J. rhombifolia and understory with high herbaceous cover. Its distribution is patchy, with patches normally small (5-10 m radius), made up of a few adult trees of J. rhombifolia, plus some Celtis tala and Scutia buxiflora trees.
Comment 6: too short introduction - I recommend you expand the first paragraph and describe and introduce the processes you will be investigating under invasion biology. This paragraph (L65-80) is misplaced, it can be moved to the begining of introduction.
Response 6: We agree. The paragraph (L65-80) was moved to the beginning of the Introduction, to expand the general presentation of the topic addressed.
Comment 7: L82-83: the questions 2 & 3 are similar and please seperate them in terms of what you investigated or rephrase based on data/methods or ...
Response 7: Question # 3 was reformulated as: 3) Does the invasibility to L. lucidum depend on the type of forest stand, defined by the dominant tree species?
Comment 8: your results must be percentages and counts in the 48 plots, and I thus, invite better statistical analysis
Response 8: The difference in density between glossy privet and the other tree species is so large and clear that, in our opinion, statistical comparison is not necessary. However, we included in the new version statistical comparisons using GLM. In any case, we used percentages in the following sentence (L113-115), which states that 99% of the individuals in the seedling assemblage are privets, and 88% in the case of saplings and poles.
Comment 9: L146: too high percentages (CV) and may mislead or means something not right . use linear mixed models to analyse the data.
Response 9: This can lead to problems when detecting patterns, and it is often suggested to use larger sampling units to reduce variance. In our case, however, we were able to find very clear spatial patterns (statistically significant), demonstrating that the evaluated predictors exert significant control over privet regeneration.
Comment 10: L 370: I am confused and stopped here. results come before methods? is this the journal style?
Response 10: We agree with the referee; it is not the usual structure. This order was stated in the template given in the journal web, for this reason we adopted it. But several recent papers published by Plants presented the standard structure. So, if it is possible for the journal, we prefer to use the classic format.
Round 3
Reviewer 3 Report
Comments and Suggestions for Authors
IMPROVE LANGUAGE
Comments on the Quality of English LanguageACCEPT